# How Broadly Neutralising Antibodies Are Redefining Immunity to Influenza

**DOI:** 10.3390/antib14010004

**Published:** 2025-01-07

**Authors:** Rebecca Steventon, Lucas Stolle, Craig Peter Thompson

**Affiliations:** Warwick Medical School, University of Warwick, Coventry CV4 7AL, UK; rebecca.j.steventon@warwick.ac.uk (R.S.); lucas.stolle@warwick.ac.uk (L.S.)

**Keywords:** influenza, virus, antibodies, broadly neutralising antibodies, vaccines, immunology

## Abstract

Recent avian influenza outbreaks have heightened global concern over viral threats with the potential to significantly impact human health. Influenza is particularly alarming due to its history of causing pandemics and zoonotic reservoirs. In response, significant progress has been made toward the development of universal influenza vaccines, largely driven by the discovery of broadly neutralising antibodies (bnAbs), which have the potential to neutralise a broad range of influenza viruses, extending beyond the traditional strain-specific response. This could lead to longer-lasting immunity, reducing the need for seasonal vaccinations, and improve preparedness for future pandemics. This review offers a comprehensive analysis of these antibodies, their application in clinical studies, and both their potential and possible shortcomings in managing future influenza outbreaks.

## 1. Introduction

Influenza is a globally endemic respiratory virus typically associated with upper respiratory tract infection, cough, and accompanying fever [1]. While generally not lethal, influenza poses a significant health burden on geriatric, paediatric, or otherwise immunocompromised individuals [2]. The World Health Organization (WHO) estimates that around one billion seasonal infections, 3–5 million cases of severe disease outcomes, and up to 650,000 annual deaths can be attributed to influenza each year [3].

Human infections are primarily caused by influenza types A and B; however, types C and D are also known. Influenza can be broadly classified by the composition of its major surface glycoproteins: the entry protein hemagglutinin (HA), and the exit protein neuraminidase (NA). The specific combination of HA and NA not only defines the virus’s preferred host target and virulence but also influences its zoonotic potential and pandemic threat [1].

Despite circulating for centuries [4], influenza remains a public health threat. The ability to continue evading existing immune responses is heavily linked to two phenomena: antigenic drift and antigenic shift. Antigenic drift describes the accumulation of glycoprotein mutations in response to selective pressure of acquired immune responses. Antigenic shift describes sudden introductions of new or recombined viral strains. The dramatic rearrangement of the antigenic landscape frequently has a devastating effect on immunologically naïve populations [5]. This has been demonstrated by the four historical flu outbreaks: the 1918 H1N1 Spanish Flu that killed an estimated 40 million people; the 1957 H2N2 Asian Flu and the 1968 H3N2 Hong Kong Flu, affecting 700,000 and 1 million people, respectively; and the 2009 H1N1 Swine Flu, affecting 16,000 people worldwide [6,7]. All except the 1918 Spanish Flu are attributed to antigenic shift, whereas the Spanish Flu is thought to have been a zoonotic avian virus that underwent unusually rapid antigenic drift [6,7].

The fact that mildly antigenically drifted seasonal strains tend to be more immunologically tolerable suggests that a person’s infection history significantly influences disease severity. In the early 1980s, distinctions were made regarding “strain-specific” and “cross-reactive” antibodies, with the latter being mentioned as a possible explanation for the ability to tolerate mildly mutated strains [8]. This was corroborated in 1993, when Okuno and colleagues observed that mice immunised with A/Okuda/57 (H2N2) gained immunity to all H1 and H2 strains through the generation of a singular broadly neutralising antibody (bnAb), termed C179 [9]. Here, we review trends and treatments relating to antibodies capable of neutralising multiple antigenically drifted, chronologically distinct viruses (intrastrain bnAbs), different viruses within the same influenza group (intrasubtypic bnAbs), and viruses within different influenza groups (intergroup bnAbs).

### Hemagglutinin

HA is the primary immunological target in influenza. Influenza A features 18 different HAs (H1–H18), while influenza B has two HAs (Yamagata and Victoria). HA is synthesised as an immature HA0 chain, which is proteolytically cleaved by endoplasmic reticulum host proteases into disulphide-linked subunits—HA1 and HA2. HA1 primarily comprises the globular head domain, forming functionally critical structures including the receptor-binding site (RBS). HA2, together with a portion of HA1, forms the stem domain. During viral infection, the HA1 RBS binds to sialic acid, inducing viral endocytosis. Upon endosomal acidification, the HA2 subunit undergoes a conformational change leading to the insertion of a hydrophobic fusion peptide into the host membrane. Alongside further conformational changes, this leads to endosomal collapse and the introduction of the viral genome to the host cell (Figure 1) [10].

Antibody potency against HA is influenced by both the mutation frequency of the epitope and its functional significance. As such, antibodies targeting the head domain, particularly the RBS, tend to be highly immunogenic; however, epitopes in the head domain are less stable and tend to drift seasonally [11]. Conversely, antibodies targeting the stem must significantly impede conformational changes during acidification of the endosome or the fusion peptide [11]. Only a few such epitopes have been characterised, yet the lower mutational rate of the HA stem means that functional antibodies are more likely to also be broadly neutralising (Figure 2) [12,13,14].

## 2. BnAbs Against the HA Stem

The discovery of C179 in 1993 demonstrated that broadly neutralising antibodies (bnAbs) targeting the hemagglutinin (HA) stem exist [9]. However, the reduced accessibility of the stem, combined with the immunodominance of the accessible HA head, has been suggested as a barrier to the development of bnAbs targeting the stem [15]. It has been shown that enhancing immune focus away from the head by hyperglycosylating variable regions significantly increases the production of stem-targeting antibodies [16]. To date, only two main immunological stem epitopes have been identified: (1) the central stem (CS) epitope, and (2) the anchor epitope/fusion peptide (Figure 2). Whether the limited identification of other stem epitopes is due to the functional importance of HA sites, steric constraints, or the dominance of other HA regions remains an open question.

Most stem antibodies characterised to date are IGVH1-69 somatically hypermutated antibodies, which predominantly bind via the heavy-chain complementarity-determining region 3 (CDRH3) (Table 1).

### 2.1. The Central Stem Epitope

Most stem bnAbs, including C179, target the central stem (CS) epitope (Figure 2 and Table 1) [17]. Different CS-targeting antibodies primarily differ in their germline-encoded IGHVs (Table 1) and specific paratopes. Mechanistically, they confer protection either by binding a hydrophobic pocket, disrupting conformational changes involved in membrane fusion and HA0 processing [18,19,20,21], or by mediating antibody-dependent cellular cytotoxicity (ADCC) (Table 1).

The first human serum-derived bnAb to the CS was discovered in 2008 [12]. This antibody, called CR6261, elicited broad protection in pandemic H5N1 and H1N1 lethally challenged mice [12]. The potency of this site became apparent with F10, a human antibody targeting the CS, capable of neutralising H1N1, H2N2, H5N1, H6N1, H6N2, H8N4, and H9N2 [19]. This was rapidly followed by the discovery of FI6, an antibody that was able to bind to all 16 hemagglutinin subtypes in influenza A, but not in influenza B [21]. FI6 has since been shown to elicit in vivo protection in mice, ferrets, and pigs against a panel of influenza A viruses (Table 1) [21,22,23].

### 2.2. The Fusion Peptide and Anchor Epitope

A second antigenic site (termed the anchor epitope) has been identified below the CS epitope and closer to the viral membrane (Figure 2 and Table 1). It was initially thought to be unique to group 2 influenza A viruses [17,24,25], with CR8020 [24] and CR8043 [25] both eliciting robust protection against H3N2 and H7Nx viruses. However, recently, group 1 influenza has also been found to contain a relevant, broadly neutralising intrastrain anchor epitope near the viral membrane [26]. This site is reported to possess a strong polyclonal response upon H1N1 vaccination or infection, with classified antibodies—047-09 4F04, 241 IgA 2F04, and 222-1C06—recognising a well-conserved epitope amongst group 1 viruses, consisting of W343, H354, Q356, S361, and Y363 [26].

**Table 1 antibodies-14-00004-t001:** Overview of extensively studied stem-targeting broadly neutralising antibodies (bnAbs): This table provides details on some of the most thoroughly researched stem-targeting bnAbs, adding to previously published work [17]. It includes information on their in vitro binding affinity, in vitro neutralisation capacity, and in vivo protective efficacy. The table also lists the immunoglobulin heavy-chain variable region (IGHV) gene used, the primary complementarity-determining region (CDR) recognition mode, and whether the antibody exhibits antibody-dependent cellular cytotoxicity (ADCC) as a significant protection mechanism. Additionally, it specifies whether the antibody was isolated from mice or humans, any known escape mutations, and the IgG subtype used in generating the findings. A“-“ is shown if the information was not provided.

	Name	In Vitro Binding	In Vitro Neutralisation	In Vivo Protection	Germline-Encoded IGHV	CDR Recognition Mode	ADCC Activity	Source	Escape Mutants	IgG Type in Studies	Ref.
Central Stem	**C179**	H1, H2, H5, H6, H9	H1, H2, H5, H6, H9	H1, H5	-	-	Yes	Mouse	T332K, V395E *	IgG2a	[9,27,28]
	**27F3**	H1, H2, H5, H6, H9, H11, H12, H13, H16, H3, H7, H10, FluB	H1, H5, H6, H3, H7, H10	-	IGHV1–69	CDRH2	-	Humans	-	IgG1	[29,30]
	**FI6**	H1–H16	H1, H5, H3, H7	H1, H5, H3	IGHV3–30	CDRH3 CDRL1	Yes	Humans	R62K, D239G, R240Q T333K, A388T °	-	[21,22,23,31,32]
	**CR6261**	H1, H2, H5, H6, H8, H9	H1, H2, H5, H6, H8, H9	H1, H5	IGHV1–69	CDRH2	Weak	Humans	A388V	IgG1	[18,30,33,34]
	**CR6323**	H1, H2, H5, H6, H8, H9	H1, H2, H5, H6, H8, H9	-	IGHV1–69	HCDR2	-	Humans	H357L/T *	IgG1	[12]
	**09-2A06**	H1	H1	-	IGHV1–69	-	-	Humans	-	-	[35]
	**09-3A01**	H1	H1	-	IGHV4–39	-	-	Humans	-	-	
	**05-2G02**	H1, H3, H5	H1, H3, H5	-	IGHV1–18	-	-	Humans	-	-	
	**A06**	H1, H5	H1, H5	H1	IGHV1–69	-	-	Humans	-	IgG1	[36]
	**39.18**	H1, H2	H1, H2	-	IGHV1–69	-	-	Humans	-	-	[37,38]
	**39.29**	H1, H2, H3	H1, H2, H3	H1, H3	IGHV3-30	CDRH3	-	Humans	G387K, D391Y/G	-	
	**81.39**	H1, H2, H3	H1, H2, H3	-	IGHV3-15	-	-	Humans	-	-	
	**36.89**	H3	H3	-	IGHV1–18	-	-	Humans	-	-	
	**FE43**	H1, H5, H6, H9	H1, H5, H6, H9	H1, H5, H6	IGHV1–69	-	-	Humans	None found	IgG1	[39]
	**FB110**	H1, H2, H5	H1, H2, H5	-	IGHV3-23	-	-	Humans	None found	IgG3	
	**3E1**	H1, H5, H9, H3, H7	H1, H5, H9, H3, H7	H1, H5	IGHV4-4	Mostly heavy chain	-	Humans	-	IgG1	[40]
	**CT149**	H1, H5, H9, H3, H7	H5, H9, H3, H7	H1, H5, H3, H7	IGHV1–18	CDRH3 CDRH2	Yes	Humans	-	IgG1	[41]
	**31.a.83**	H1, H2, H5, H9, H3, H7	H1, H2, H5, H9, H3, H7	-	IGHV3–23	Mostly CDRH3 CDRH2	-	Humans	-	-	[42]
	**56.a.09**	H1, H5, H3, H7	H1, H5, H3, H7	-	IGHV6–1	Mostly CDRH3 CDRH2	-	Humans	-	-	
	**CR9114**	H1, H2, H5, H6, H8, H9, H12, H13, H16, H3, H4, H7, H10, H15, FluB	H1, H2, H5, H6, H8, H9, H12, H3, H4, H7, H10	H1, H2, H3, H5, H9, FluB	IGHV1–69	CDRH2	Weak	Humans	R62K, D239G, R240Q, L335V, D363G, A388T °	IgG1	[30,31,33,43,44]
	**F10**	H1, H2, H5, H6, H8, H9, H11, H13, H16	H1, H2, H5, H6, H8, H9, H11	H1, H5	IGHV1–69	CDRH2	Yes	Humans	N460, S123, E190D+G225D, N203VHA + E329KNA *	IgG1	[19,30,32,45]
	**MEDI8852**	H1–H18	H1, H2, H5, H6, H9, H3, H7	H1, H5, H3	IGHV6-1	CDRH2 CDRH3 CDRL1	Yes	Humans	-	IgG1	[46,47]
	**CR9117**	Mouse homologue of CR9114, presumed to have similar neutralisation capacity			-		Yes	Mouse	-	IgG2a	[33]
Anchor Domain	**Polyclonal response (FISW84/222-1C06 were named)**	H1, H2, H5	H1, H2, H5	H1	IGHV3-23 IGHV3-30 IGHV3-30-3 IGHV3-48	CDRk3 CDRH2 CDRH3	No	Humans	-	IgG1	[26]
Fusion Peptide	**CR8020**	H3, H4, H7, H10, H14, H15	H3, H7, H10	H3, H7	IGHV1–18	CDRH1 CDRH3	Weak	Humans	D372N, G376E *	IgG1	[20,25,48,49]
	**CR8043**	H3, H4, H7, H10, H14, H15	H3, H7, H10	H3, H7	IGHV1–3	CDRH1 CDRH3	-	Humans	R378M, Q380R/T *	IgG1	[25,49]
	**9H10**	H3, H9	H3, H10	H3	-	-	-	Mice	R378M T385R Q387R/T G386E *	-	[49]

Note: ° = numbering from methionine; * = H3 numbering.

## 3. BnAbs Against the HA Head Domain

The low mutational rate of the HA stem has historically made it a more heavily investigated vaccine and bnMab therapy target, yet this usually came with the trade-off of lower-potency Mabs [11] and sterically occluded antigenic sites [50], where the stem’s target regions are physically shielded or less accessible due to surrounding viral structures. In contrast, the head domain is mutationally volatile and immunodominant, yielding a more potent and diverse set of antibodies [15] with respect to both their germline sequences and their binding mechanisms (Table 2).

Many bnAbs against the head have been characterised, and their epitopes can be categorised into four distinct sites: the receptor-binding site (RBS), the lateral patch, the vestigial esterase (VE), and the occluded site (Figure 2).

### 3.1. Receptor-Binding Site

Broadly neutralising antibodies against the receptor-binding site (RBS) have been described in the literature for almost as long as stem antibodies (Table 2 and Figure 2) [51]. These are usually characterised by hemagglutination inhibition (HAI), as antibodies that directly compete with sialic acid for the RBS [51,52,53,54].

Antibodies that neutralise the RBS frequently do so through molecular mimicry, sterically and electrostatically mimicking sialic acid [53,55]. Curiously, while the germline-encoded IGHVs differ widely between different RBS bnAbs, the dominant loop associated with RBS binding tends to be CDRH3. This CDRH3-centric mechanism makes RBS antibodies particularly susceptible to mutations in and around the RBS, as small steric or electrostatic constraints in the CDR insertion path can completely abolish antibody binding [30,39,53,56,57,58].

### 3.2. Lateral Patch

The lateral patch is a region on the HA head that is offset from the RBS (Figure 2). However, reports on HAI activity [59] indicate active or passive inhibition of sialic acid binding. The original study that coined the term “lateral patch” characterised CL6649, an antibody with H1N1 intrastrain activity. However, mechanistic insight into the mode of neutralisation was lacking [59,60].

A recent study on H7N9 lateral patch antibodies may offer further insights on the mechanisms involved with this site. Jia and colleagues found that sialic acid binding is passively inhibited by the antibody H7.HK1. This antibody inhibits the HA 220 loop (G218–G228 in H7 numbering, or G228-238 in H3 numbering), which makes hydrophobic contacts with sialic acid [61]. It is worth noting, however, that these mechanisms may not translate between influenza subtypes, and further research is needed to understand how CL6649 and H7.HK1 compare.

### 3.3. Vestigial Esterase

The vestigial esterase (VE) is a region located in the HA head between the RBS and the start of the HA stem (Figure 2) [62]. Its sequences are highly conserved within subtypes, but not across subtypes [61,62]. VE-specific antibodies tend to lack HAI activity, as the viral RBS remains free to bind to sialic acid and hemagglutinate cells [63,64]. Instead, these antibodies primarily elicit protection via ADCC through Fc-FcγR responses [61,62,64,65,66], and they may be involved in crosslinking, thereby conformationally restricting different HA trimers [63]. One characterised VE-specific antibody, 46B8, has additionally been reported to impair membrane fusion by blocking the conformational change from the prefusion to the extended intermediate state [66] (Figure 1a,b). However, even for 46B8, there is no HAI activity, and Fc-FcγR ligation is thought to be one of the primary protective pathways [66].

VE-specific bnAbs have been described against group 1 [62,67], group 2 [63,68], and influenza B [54,66], with ADCC consistently being described as a crucial mechanism for in vivo protection (Table 2).

### 3.4. Interface and Occluded Epitope

Finally, the occluded epitope is a name given to an epitope either sitting between two HA monomers or within the HA core. Despite being characterised as early as 1993 [69], the value of this low-variability site for bnAbs had not been realised until three independent research teams demonstrated its ability to provide therapeutic protection against a range of different influenza A viruses, including H1N1, H3N2, H5N1, and/or H7N9 [70,71,72].

Despite its occluded nature, this epitope is available for binding even in intact, trimeric HAs [72], likely through protein dynamics. However, the mechanism by which protection is achieved has been reported to differ widely.

FluA-20 is a bnAb that has been shown to protect mice against viral challenge with H1N1, H3N2, H5N1, or H7N9 subtypes. It has been found to bind the occluded epitope in uncleaved immature HA0. Trypsin-based treatment destabilises the FluA-20 contacts, implying that FluA-20 may disrupt HA trimerisation. No HAI activity or significant ADCC has been observed [70].

Conversely, the 8H10 bnAb has been shown to bind mature trimeric HAs and elicit protection against various H3N2 strains, as well as to bind to H4 viruses. Protection is thought to be primarily governed by ADCC [72]. These findings are corroborated in another study that showed a panel of antibodies targeting the occluded epitope to be eliciting protection primarily through ADCC [71].

It is worth noting that broadly neutralising sites have also been reported at an exposed site of the HA multimerisation interface (Figure 2) [73]. This bnAb, termed F005-126, lacks HAI activity and protects HA from trypsin-based cleavage against a panel of H3N2 viruses, indicating inhibition of endosomal rearrangements [73]. 

**Table 2 antibodies-14-00004-t002:** Overview of extensively studied head-targeting broadly neutralising antibodies (bnAbs): This table provides details on some of the most thoroughly researched head-targeting bnAbs, adding to previously published work [17]. It includes information on their in vitro binding affinity, in vitro neutralisation capacity, and in vivo protective efficacy. The table also lists the immunoglobulin heavy-chain variable region (IGHV) gene used, the primary complementarity-determining region (CDR) recognition mode, and whether the antibody exhibits antibody-dependent cellular cytotoxicity (ADCC) as a significant protection mechanism. Additionally, it specifies whether the antibody was isolated from mice or humans, any known escape mutations, and the IgG subtype used in generating the findings. A “-“ is shown if the information was not provided.

	Name	In Vitro Binding	In Vitro Neutralisation	In Vivo Protection	Germline-Encoded IGHV	CDR Recognition Mode	ADCC Activity	Source	Escape Mutants	IgG Type in Studies	Ref.
RBS	**S139/1**	H1, H2, H3, H5, H9, H13	H1, H2, H3, H5, H9, H13, H16	H1, H3	-	CDRH2	-	Mouse	K156, G158, S193, insertion at 133a *	IgG2a	[52,53,74]
	**C05**	H1, H2, H9, H12, H3	H1, H2, H3	H1, H3	IGHV3-23	CDRH3	Weak	Human	insertion at 133a	-	[53,75]
	**F045-092**	H1, H2, H13, H3	H1, H2, H3, H13	-	IGHV1–69	CDRH3	-	Human	133A insertion *	-	[30,56]
	**K03.12**	H1, H3	-	-	IGHV1-2	CDRH3	-	Human	-	IgG1	[76]
	**2G1**	H2	H2	H2	IGHV1–69	-	-	Human	-	-	[30,77]
	**FE17**	H1, H9	H1, H9	H1, H5	IGHV1–69	-	-	Human	S145N *	IgG1	[39]
	**12H5**	H1, H5	H1, H5	H1, H5	IGHV9-1 alignment	CDRH2, CDRH3	-	Mouse	Y98A, A137E, H141A, A142E, G143R, A144E, W153A, D190A *	IgG1	[58]
	**1F1**	H1	H1	H1	IGHV3-30	CDRH3	-	Human	D190E, D225G *	-	[78]
	**5J8**	H1	H1	H1	IGHV4-b	-	-	Human	R(133A)I, K(133A)Q, A137T, D199H, K222Q *	-	[57]
	**CH65**	H1	H1	-	IGHV1-2	CDRH3	-	Human	G200D, K/R insertion at 133A *	IgG1	[55,79]
	**CH67**	H1	H1	-	IGHV1-2	CDRH3	-	Human	likely as CH65	IgG1	[55,79]
	**3D11**	H1	H1	H1	-	-	-	Mouse	K153E, D200E *	IgG1	[80]
	**8M2**	H2	H2	H2	IGHV1–69	-	-	Human	G142D *	-	[30,77]
	**8F8**	H2	H2	H2	IGHV3-33	-	-	Human	R144Q/M/K, T134K *	-	[77]
	**A2.91.3**	H3	H3	-	-	CDRH3	-	Mouse	K189N, F193S/K, L194P, Y195A *	IgG1	[81]
	**AVFluIgG03**	H5	H5	H5	IGHV3-23	CDRH3	-	Human	S159I, R193M/W *	IgG1	[82,83]
	**FLD21.140**	H5	H5	H5	IGHV4-31	CDRH3	-	Human	S159I, R193M/W *	IgG1	[83]
	**13D4**	H5	H5	H5	Mouse IGHV1-9	CDRH3	-	Mouse	K/R193N *	-	[82]
	**Hab21**	H5	H5	-	-	-	-	Mouse	H136A, D197A, A198G, A199G, E200A, N207A, P208A, P225A, N258A *		[84]
	**H5.3**	H5	H5	-	-	CDRH3	-	Human	-	-	[85]
	**CR8033**	FluB	FluB	FluB	IGHV3-9	CDRH dominant	-	Human	P161Q *	-	[54]
VE	**H3v-47**	H3	H3	H3	IGHV1–69	CDRH2, CDRH3, CDRL1, CDRL3	Yes	Human	None found	IgG1	[63]
	**F005-126**	H3	H3	-	-	CDRH3	-	Human	N285Y *	IgG1	[73]
	**H5M9**	H5	H5	H5	-	CDRH1-3, CDRL1-2	-	Mouse	D53A/N, E78K, E83aA/K, Y274A *	IgG1	[86]
	**9F4**	H5	H5	H5	-	-	Yes	Mouse	R62G *	IgG2b	[87,88]
	**100F4**	H5	H5	H5	-	-	Yes	Human	D72A, E116Q/L *	-	[89,90]
	**4F5**	H5	H5	H5	IGHV3-43	-	Yes	Human	W70, L71, L72, G73, N74, P75 *	-	[91,92]
	**1H5**	H7	-	H7	-	-	Yes	Mouse	R58K *	IgG2a	[93]
	**1H10**	H7	-	H7	-	-	Yes	Mouse	R58K *	IgG2a	[93]
	**46B8**	FluB	FluB	FluB	-	-	Yes	Human	S301F *	IgG1	[66]
	**CR8071**	FluB	FluB	FluB	IGHV1-18	-	Yes	Human	None found	-	[54,66]
Lateral Patch	**CL6649**	H1	H1	-	IGHV4-39	CDRH3, CDRL1, CDRL3	-	Human	K176Q S175N+K176Q *	-	[60]
	**H7.HK1**	H7, H10, H15	H7	H7	IGHV4-59	CDRH1-3, CDRL1, CDRL2	-	Human	R57K *	IgG1	[61,94]
	**07-5F01**	H7	H7	H7	IGHV4-31	-	-	Human	R57K *	IgG2a	[61,94]
HA Multimerisation Interface and Occluded Site	**FluA-20**	H1-H12, H14, H15	-	H1, H3, H5, H7	IGHV4-61	CDRH3, CDRL2	Weak	Human	In H1: P103G, R230A, P231G, V233G, R239A *	-	[70]
	**8H10**	H3, H4	H3	H3	IGHV5-9-1	CDRH1-3, CDRL1, CDRL3	Yes	Human	-	IgG2a, IgG1	[72]
	**S5V2-29**	H1, H2, H3, H4, H7, H9, H14	-	H1, H3	IGHV4-61	-	Yes in IgG2c but not IgG1	Human	-	IgG1 and IgG2c	[71]
	**H2214**	H1, H2, H3, H4, H14	-	H1, H3	IGHV3-23	-	Yes in IgG2c but not IgG1	Human	-	IgG1 and IgG2c	[71]
	**H7-200**	H7, H15	-	H7	-	CDRH dominant, CDRL3	-	Human	-	-	[95]
	**H7.5**	H7	H7	-	-	CDRH2, CDRL3	-	Human	-	-	[95,96]

Note: * = H3 numbering.

## 4. BnAbs in Clinical Trials

In the 1970s, Köhler and Milstein pioneered hybridoma technology, facilitating the production of monoclonal antibodies [97]. This breakthrough has significantly advanced the therapeutic use of monoclonal antibodies, particularly in autoimmune diseases, oncology, and infectious diseases. Broadly neutralising monoclonal antibodies (bnMAbs) hold promise as a therapeutic alternative to existing influenza treatments, particularly in combating influenza A. Ongoing clinical trials aim to evaluate the efficacy and safety of stem-targeting bnMAbs in treating influenza infections (Table 3).

### 4.1. CT-P27

CT-P27 is a combination therapy of two bnMAbs that target the stem region of group 1 and group 2 viruses, and it showed a reduction in the mean area under the curve (AUC) of viral load when compared to a placebo (NCT02071914). During a phase 2a clinical trial, 61 healthy volunteers were given a single IV infusion of either 10 mg/kg of CT-P27, 20 mg/kg of CT-P27, or a placebo before challenge with an influenza A virus. Groups who received an infusion of CT-P27 experienced a reduced mean AUC of viral load when measured by quantitative PCR of nasopharyngeal swabs over a 9-day period (NCT02071914). However, a further clinical trial, NCT03511066, to study the effects of CT-P27 in patients with acute, uncomplicated influenza A, was terminated early due to the inactivation of CT-P27.

### 4.2. MED18852

MED18852 is an IgG1 bnMAb that has been shown to target multiple influenza A viruses via the stem region [46] and was shown to cause no statistical improvement in patients’ disease outcomes when compared to current treatments (NCT02603952). During a phase 1/2a clinical trial, 126 participants aged 18–65 suffering with acute, uncomplicated influenza A infection were given either a single intravenous infusion of 750 mg of MED18852 or 3000 mg of MED18852 followed by a 5-day course of oseltamivir, a placebo plus a 5-day course of oseltamivir, or just a 3000 mg MED18852 infusion. No reduction in symptoms was seen over the 13-day period for patients receiving MED18852, along with no change in viral shedding. An increase in treatment-emergent adverse events (TEAEs) was seen in patients who received MED18852 compared to the placebo group (NCT0203952). A further clinical trial to assess high and low doses of MED18852 in conjunction with oseltamivir, compared to a placebo with oseltamivir, was withdrawn due to the company’s decision.

### 4.3. VIS410

VIS410 is an IgG1 bnMAb that has been engineered to bind to the stem region of group 1 and group 2 viruses and has been shown to statistically improve the signs and symptoms of influenza infection and reduce the time to resolution of peak viral load when compared to a placebo, but it does not provide a statistically significant improvement over current treatments (NCT02989194, NCT03040141). In a phase 2a clinical trial in 2016, following challenge with H1N1 virus, a single IV infusion of either 2300 mg or 4600 mg of VIS410 was given to healthy volunteers aged between 18 and 45 years [98]. The viral load AUC was compared to a placebo group, and a significant reduction in viral load AUC when compared to the placebo was seen within the study, with no increase in clinical symptoms [98]. A second phase 2 clinical trial looked at the effects of VIS410 in patients with acute, uncomplicated influenza infection; 150 participants aged 18–65 were recruited and given either a low dose of 2000 mg of VIS410, a high dose of 4000 mg of VIS410, or a placebo. Both doses of VIS410 showed a statistically significant improvement in patient-reported signs and symptoms of influenza infection on days 3 and 4, while also showing a statistically significant reduction in the time to resolution of peak viral load when compared to the placebo (NCT02989194). A final phase 2 clinical trial compared the effects of low- or high-dose VIS410 combined with oseltamivir to those of oseltamivir alone in severely ill patients with influenza infection requiring oxygen support (NCT03040141). A total of 75 patients, with a mean age of 61, were given either a single IV infusion of 2000 mg of VIS410 plus oral oseltamivir, 4000 mg of oral oseltamivir, or a placebo plus oral oseltamivir. No statistically significant reduction in viral load or cessation of oxygen was seen in patients who received VIS410 compared to the placebo group.

### 4.4. MHAA4549A

MHAA4549A is an IgG1 bnMAb that can target multiple subtypes of influenza and has been shown to reduce the AUC of the H3 virus in a challenge model, as well as reducing the number of days to alleviate symptoms in patients with uncomplicated influenza A infection, but did not improve clinical outcomes when compared to current standard treatments (NCT01980966, NCT02293863, NCT02623322). In a phase 2 clinical trial, 100 healthy volunteers were inoculated with H3N2 before receiving a single IV infusion of either placebo or 400 mg, 1200 mg, or 3600 mg of MHAA4549A (NCT01980966). No treatment effects were seen with the 1200 mg group, but both the 400 mg and 3600 mg groups saw a reduction in the AUC of viral load, with the greatest reduction seen in the 3600 mg group, along with a reduction in the duration of viral shedding [99]. Both the 400 mg and 3600 mg groups also saw statistically significant reductions in total mucus weight compared to the placebo group, along with reduced cytokine levels [99]. A second phase 2 clinical trial was carried out with patients hospitalised with severe influenza A. A total of 127 patients, with a mean age of 60, received either a placebo IV infusion, a 3600 mg IV infusion of MHAA4549A, or a 8400 mg IV infusion of MHAA4549A, all followed by standard oseltamivir treatment for 5 days (NCT02293863). MHAA4549A did not statistically reduce the time to normalisation of respiratory function compared to the placebo, nor did it reduce the clinical resolution of symptoms or the peak viral load compared to oseltamivir alone (NCT02293863). A final phase 2 trial was carried out in patients with uncomplicated seasonal influenza A infection. A total of 120 patients, with a mean age of 37, received with a single IV infusion of placebo, 3600 mg of MHAA4549A, or 8400 mg of MHAA4549A (NCT02623322). Patients receiving the 3600 mg dose of MHAA4549A had a statistically significant reduction in the time required to reduce the symptom score to below 7, compared to the controls. However, there was no statistically significant reduction in patients experiencing complications of influenza (NCT02623322).

### 4.5. CR8020 and CR6261

CR8020 and CR6261 are bnMAbs that target group 1 and group 2 influenza. CR6261 was able to statistically reduce symptoms in an H1N1 challenge study, and a joint clinical study looking at CR8020 and CR6261 was withdrawn (NCT02371668, NCT01992276). In a phase 2 clinical trial, 20 participants, with a mean age of 26, received either an IV infusion of placebo or 15 mg/kg CR8020 (2013-002185-39). Unfortunately, the results from that study cannot be considered, as 0% of the participants in the placebo group had a detectable and quantitative influenza infection rate (2013-002185-39). A phase 2 clinical trial looking at CR6261 studied 91 participants between 18 and 45 years old within an H1N1 challenge model. Participants were given either a placebo or 50 mg/kg CR6261 as an IV infusion (NCT02371668). CR6261 was able to statistically reduce the percentage of participants who experienced symptoms but did not reduce the number of participants who experienced mild-to-moderate disease. Viral shedding and days of shedding were not reduced for participants who received CR6261 compared to the placebo (NCT02371668). A joint clinical trial of CR8020 and CR6261 was withdrawn due to preliminary efficacy results from the influenza challenge studies (NCT01992276).

### 4.6. Future Directions for Therapeutic bnMAbs

Initial results from clinical trials suggest that bnMAbs may provide significant clinical benefits in managing uncomplicated influenza A, as seen with the bnMAbs VIS410 and MHAA4549A However, when compared to the current treatment option of oral oseltamivir, no bnMAb in present clinical trials was able to provide a reduction in symptoms or viral load. Of the bnMAbs in current clinical trials, three have had studies withdrawn due to either company decisions or lack of efficacy of the bnMAb. These findings emphasise the potential of bnMAbs as a therapeutic strategy against uncomplicated influenza infection, while also underscoring their limitations and the need for further research.

Current clinical trials for influenza predominantly utilise IgG1 bnMAbs. IgG1 and IgG3 are the major subclasses generated during viral infections, with distinct functional differences and characteristics. IgG1 has a shorter 15-amino-acid hinge region with only two disulphide bonds, providing a longer half-life and potentially greater therapeutic efficacy [100,101]. In contrast, IgG3, with its shorter half-life, has been overlooked for therapeutic applications. However, with its longer hinge region of 62 amino acids and 11 disulphide bonds [100,101], IgG3 may be able to overcome steric hindrance, a limitation observed in bnAbs targeting the fusion peptide and anchor epitope [20]. Tharakaraman and colleagues found that the viral membrane could sterically hinder the interaction of CR8020, a fusion peptide antibody, with the two most common Fc receptors in humans when presented as an IgG1 antibody [20]. The longer hinge region of IgG3 would allow the Fc region of the antibody to be further removed from the viral membrane, potentially removing the steric hindrance and allowing greater engagement of the Fc receptors by the antibody.

Moreover, IgG3 has a higher affinity for the Fc receptors FcγRIIa, FcγRIIIa, and FcγRIIIb in its monomeric form compared to IgG1, while its binding efficiency in its complex form exceeds that of IgG1 for all receptors [100], making it particularly effective at activating complement-dependent cytotoxicity (CDC) and ADCC [101], mechanisms employed by multiple head- and stem-targeting bnAbs (Table 1 and Table 2) to indirectly neutralise influenza [102,103,104]. DiLillo and colleagues found that activation of ADCC was vital for the in vivo efficacy of certain stem-targeting bnAbs [105]. By switching the subclass of bnAbs from an IgG1 to an IgG3, the efficacy of stem-targeting antibodies could be enhanced, improving their therapeutic potential. Switching the subclass of MAbs has been seen to have beneficial effects for SARS-CoV-2, with the switch from an IgG1 to an IgG3 enhancing both Fc-mediated phagocytosis and the triggering of the classical complement pathway [106]. Additionally, Bolton and colleagues found that IgG3 antibodies exhibited superior binding and neutralisation capacity to antigenically drifted influenza and SARS-CoV-2 viruses relative to other IgG subclasses [107].

To enhance the therapeutic efficacy of IgG1 bnMAbs, modifications to the Fc region could be explored to improve ADCC initiation. Ocaratuzumab is an anti-CD20 mAb that has the Fc modifications P247I/A339Q, which have been shown to increase its binding to lower-affinity FcγRIIIa, allowing it to have increased ADCC activity [108,109,110]. Fc modifications can also be used to increase the half-life of antibodies to improve their therapeutic efficacy. The MAb therapy sotrovimab, which was approved for use against SARS-CoV-2, has the Fc modifications M428L/N434S to extend the half-life of the antibody [111].

Another promising approach involves the development of chimeric IgG1/IgG3 antibodies, which may enhance ADCC and CDC as well as improving binding to sterically hindered stem regions. Natsume and colleagues generated a chimeric form of rituximab (an anti-CD20 antibody) that consisted of the CH1 and hinge regions from IgG1, with the Fc region of IgG3 and the COOH terminal CH3 domain of IgG1. This chimeric antibody showed enhanced CDC and ADCC activities compared to the wild type [112]. Chimeric antibodies could combine the favourable pharmacokinetics of IgG1 with the functional advantages of IgG3, potentially overcoming the limitations associated with IgG3 monotherapy [112,113]. These strategies highlight avenues for optimising bnMAb design to improve their utility in influenza treatment. Further research into IgG subclass-specific characteristics and their impact on bnMAbs’ efficacy is warranted.

**Table 3 antibodies-14-00004-t003:** Table of stem-targeting bnMAbs currently undergoing clinical trials for use in influenza infection: Information about the type of antibody, target, dose regime, and results was taken from the corresponding study record listed on the NIH clinical studies site.

Name	Type and Target	Dosage/Infection Model	Result	Trial Registry ID/Reference
CT-P27	CT-120 and CT-149 mAbs targeting the stem region of group 1 and group 2 influenza hemagglutinin	10 mg/kg CT-P27, 20 mg/kg CT-P27, or placebo in an influenza challenge model	Reduction in AUC of viral load, as measured by quantitative PCR of nasopharyngeal swabs for patients who received CT-P27	NCT02071914, [102]
		90 mg/kg CT-P27, 45 mg/kg CT-P27, or placebo	NCT03511066 was terminated due to CT-P27 inactivation	NCT03511066.
MEDI8852	Human IgG1 kappa monoclonal antibody (MAb) targeting H1N1 and H3N2 viruses, as well as subtypes such as H2, H5, H6, H7, and H9 via the stem region	750 mg or 3000 mg of MEDI8852 given with oseltamivir, or 3000 mg of MEDI8852 alone, to patients with acute, uncomplicated influenza caused by type A strains	MEDI8852 provided no statistically significant improvement over oseltamivir alone, and it potentially worsened disease in combination compared to oseltamivir alone	NCT02603952, [46]
		Low and high doses of MEDI8852 and oseltamivir in comparison to oseltamivir and placebo	Withdrawn due to company decision	NCT03028909
VIS410	Human immunoglobulin IgG1 monoclonal antibody engineered to bind to the stem region of group 1 and 2 influenza A hemagglutinins	Influenza challenge with H1N1 followed by a single administration of VIS410 or placebo	No results posted	NCT02468115, [98]
		2000 mg or 4000 mg of VIS410 was given to patients with uncomplicated influenza A infection and compared to a placebo	Statistically significant improvements in signs and symptoms of influenza infection on days 3 and 4 with VIS410 compared to placebo; statistically significant reduction in time to resolution of peak viral load when patients were given VIS410	NCT02989194, [114]
		3600 mg or 8400 mg of VIS410 combined with oral oseltamivir, or placebo with oseltamivir, in patients hospitalised with influenza A infection	No statistically significant reduction in time to cessation of oxygen, or reduction in viral load, in nasopharyngeal samples	NCT03040141
MHAA4549A	Human monoclonal antibody, IgG1, targeting the influenza A virus hemagglutinin stem across multiple subtypes	Influenza challenge with H3N2 influenza virus followed by a dose of 400 mg, 1200 mg, or 3600 mg	No results posted	NCT01980966, [99]
		3600 mg or 8400 mg given either on its own or with oseltamivir to patients hospitalised with severe influenza infection	MHAA4549A did not improve clinical outcomes over OTV alone; MHAA4549A + OTV did not further reduce viral load versus placebo + OTV; MHAA4549A did not alleviate symptoms quicker than a placebo	NCT02293863, [115]
		3600 mg or 8400 mg given to patients with uncomplicated seasonal influenza A infection	The 3600 mg dose was able to statistically reduce the time required to reduce the symptom score to below 7 compared to the control	NCT02623322
CR8020	An mAb targeting the stem region of group 2 influenza A hemagglutinin	15 mg/kg CR8020 given before challenge with an H3N2 influenza virus	No results	NCT01938352
CR6261	An mAb that targets the stem region of group 1 and group 2 influenza hemagglutinin	50 mg/kg administered one day after challenge with H1N1	Statistically reduced percentage of participants who experienced influenza symptoms; no statistically significant reduction in AUC or viral shedding.	NCT02371668, [116]
CR8020/CR6261			Withdrawn due to preliminary efficacy results from an influenza challenge trial	NCT01992276

## 5. Broadly Protective Vaccines in Clinical Trials

BnMAbs represent a promising secondary defence mechanism against influenza, complementing the primary defence provided by vaccination. Current influenza vaccines require annual updates to address antigenic drift and shift. These processes result in a constantly changing virus population, necessitating frequent reformulations of vaccines to match circulating strains, and sometimes leading to mismatched vaccines with reduced efficacy. However, the induction of bnAbs via vaccination could offer cross-protection against multiple strains of influenza, irrespective of antigenic variations. Vaccines designed to elicit bnAbs may provide long-term immunity and reduce the need for frequent vaccine reformulations. Several clinical trials are currently investigating novel influenza vaccines aimed at inducing bnAb formation (Table 4). These trials employ various strategies to stimulate bnAb production specifically targeting the hemagglutinin protein of influenza A.

### 5.1. UFluA

One promising approach involves presenting only the stem region of the influenza virus, aiming to elicit bnAbs against conserved stem epitopes across diverse influenza strains. UFluA, a stabilised stem nanoparticle vaccine currently in phase 1 trials (NCT05155319), exemplifies this strategy. By stabilising the HA stem into a nanoparticle format, UFluA aims to induce bnAbs that are effective against both group 1 and group 2 influenza A viruses, offering broad cross-protection.

### 5.2. H1ssF

Another vaccine candidate, H1ssF, employs the stem domain from influenza A/New Caledonia/20/1999 (H1N1), genetically fused to the ferritin protein from *Helicobacter pylori*. This design is intended to enhance the presentation of the stem region to the immune system, thereby inducing bnAbs targeting the stem. Initial results from a phase 1 trial demonstrated that H1ssF generated an increased IC80 concentration in a pseudoviral neutralisation assay against the homologous H1N1 A/New Caledonia/20/99 virus (NCT03814720), indicating promising immunogenicity.

### 5.3. G1 mHA

The G1 mHA vaccine utilises a “mini-HA”, a stabilised form of the HA stem trimer [117]. Previous studies have shown that this design can induce stem-targeting antibodies against various group 1 viruses in non-human primates [118]. Currently, G1 mHA is undergoing a phase 1/2 trial (NCT05901636) to further evaluate its efficacy and safety in humans.

### 5.4. GSK3816302A

Chimeric vaccines represent another innovative approach to induce stem-targeting bnAbs. These vaccines employ a prime–boost regimen, using vaccines with different HA head domains but a consistent stem domain to focus the immune response on the stem. GSK3816302A, a chimeric vaccine currently in phase 1 trials (NCT03275389), incorporates cH8/1 N1, cH5/1 N1, and cH11/1 N1 constructs to elicit bnAbs targeting the H1 stem. Initial results indicate an increase in anti-H1 stem antibodies post-vaccination, with a statistically significant humoral immune response. Notably, increased antibody titres against the H2 and H18 subtypes were also observed, suggesting potential cross-reactivity [119].

### 5.5. fH1/DSP-0546LP

A specific subset of bnAbs target the HA stem region in its post-fusion form (Table 1). To induce these bnAbs, vaccines must present the stem in a non-native post-fusion conformation. Previous studies have demonstrated that vaccines designed to present this form can induce protective bnAbs in mice, offering cross-protection against mismatched influenza A strains [120]. A clinical trial evaluating a post-fusion hemagglutinin antigen is currently in the recruitment phase (NCT06460064).

### 5.6. M-001

In parallel, a universal influenza vaccine, M-001, which incorporates conserved epitopes from the M1 matrix protein, NP, and HA of both influenza A and B, has advanced to phase 3 trials [121]. Despite initial promise, results from this trial indicated no statistically significant differences between the control and vaccine groups regarding the prevention of influenza illness or reduction in symptom severity (NCT03450915). These results show that, while inducing the production of bnAbs against hemagglutinin is an attractive route forward for the formulation of a universal influenza vaccine, many challenges exist for these vaccines, and continued research into this area is required.

**Table 4 antibodies-14-00004-t004:** Table of broadly protective vaccines and vaccine antigens targeting the hemagglutinin currently undergoing clinical trials for use in influenza prevention.

Phase	Name of Vaccine	Target/Type of Vaccine	Dosage/Infection Model	Results	Trial Registry ID/Reference
**Recruiting**	fH1/DSP-0546LP	Post-fusion hemagglutinin antigen	Combination of 2 dose levels of fH1 (2 and 8 μg), 3 dose levels of DSP-0546LP (2.5, 5, and 10 μg), and placebo; each dose level of fH1 will be combined with the low, medium, and high doses of DSP-0546LP to assess safety, tolerability, and immunogenicity	Active	NCT06460064, [120]
**Phase 1**	EBS-UFV-001	Induction of antibodies against conserved stem antigens across group 1 and 2 via a hemagglutinin-stabilised stem nanoparticle vaccine	Testing the safety, tolerability, and immunogenicity of 20 µg or 60 µg of UFluA as single dose or as two doses	No results posted	NCT05155319, [122]
	H1ssF	HA stem domain from influenza A/New Caledonia/20/1999 (H1N1), genetically fused to the ferritin protein from *H. pylori*	20 mcg was given to group 1; group 2 received 60 mcg on a prime–boost schedule	All regimes generated an increased IC80 concentration when tested in a pseudoviral neutralisation assay against the homologous H1N1 A/New Caledonia/20/99 virus	NCT03814720
	GSK3816302A	Chimeric vaccines of D-SUIV cH8/1 N1, D-SUIV cH5/1 N1, and D-SUIV cH11/1 N1 to induce cross-reactive stem-targeting antibodies against the H1 stem	Chimeric H5, H8, and H11, with and without the adjuvants AS03 or AS01, were tested for their reactogenicity, safety, and immunogenicity; H8 and H5 were given with a placebo second dose, or all three were given	An increase in anti-H1 stem antibodies, as measured by ELISA and MN assay, was seen across all dose schedules, with the adjuvant AS03 providing a statistically significant increase in humoral immune response for anti-H1 stem antibodies by ELISA at day 29 and day 85; increases in antibody titres against H2 and H18 were also identified	NCT03275389, [119]
**Phase 1/2**	G1 mHA	Mini-hemagglutinin stem-derived protein vaccine antigen	Single dose of influenza G1 mHA with or without Al(OH)_3_ adjuvant at two dose levels to evaluate safety, reactogenicity, and immunogenicity	Active	NCT05901636, [117,118]
**Phase 3**	(M-001)	A recombinant 45 kDa protein produced in *Escherichia coli*, consisting of three repetitions of nine linear, conserved influenza A and B epitopes to form a single recombinant protein; epitopes were derived from M1 matrix protein, NP, and HA	Vaccination with a 1 mg dose of M-001 twice: once at day 0 and once at day 21, and then followed for 2 years	No statistical difference in prevention of influenza infection; did not statistically reduce the number of patients with influenza-like symptoms or the severity of either qRT-PCR- or culture-confirmed influenza illness	NCT03450915, [121]

## 6. Limitations of bnAbs Within Influenza Therapeutics and Vaccines

BnAbs present a promising avenue for the development of universal influenza vaccines and therapeutic interventions for influenza infections. However, bnAbs do have potential downfalls within these areas.

Although several bnAbs have been identified, no bnAb thus far possesses complete universality (Table 1 and Table 2). This limitation necessitates precise identification of the influenza strain prior to bnAb administration, which is not a common practice in clinical settings, reducing their practicality for routine treatment.

### 6.1. Escape Mutations

Influenza undergoes antigenic drift and shift over time, resulting in mutations within the HA protein. These mutations can restrict the effectiveness of certain bnAbs. For instance, an amino acid insertion at position 133 induces a localised structural change in the 130 loop of the HA head domain. This insertion is observed in 100% of all H6 and H10, ~95% of all H5, and ~70% of all H1 sequences. This alteration inhibits the activity of multiple RBS-targeting antibodies (Table 2), such as S139/1 and C05, as demonstrated by Ekiert and colleagues, as well as by Lee and colleagues [53,74]. As a result, these antibodies may have reduced efficacy as therapeutic agents. Furthermore, the antibody FluA-20, which targets an epitope within the HA trimer interface, has already encountered escape mutations (Table 2). A mutation from an arginine to an isoleucine at position 229, one of the five major epitope residues, has already been seen in an H3 strain and rendered FluA-20 ineffective [70].

Although the stem region of HA is less prone to antigenic drift and shift, therefore being more conserved than the head region, escape mutations in stem-targeting bnAbs have also been identified (Table 1). Ekiert and colleagues identified two escape mutations for the fusion-peptide-targeting antibody CR8020 [24], which had entered clinical trials (Table 3), while Okuno and colleagues identified two escape mutations for the central stem-targeting antibody C179 [9]. Although these mutations have not yet been seen in naturally occurring viruses, the potential for viral escape from stem-targeting antibodies is possible.

A further concern is the potential for vaccines and therapeutic bnAbs that target regions susceptible to escape mutations to drive antigenic drift, leading to the emergence of strains that are resistant to such therapies. The generation of bnAbs through universal vaccines, or the therapeutic use of bnAbs, could potentially induce mutations in currently conserved regions. Chai and colleagues determined that although Ser301, a key epitope residue at the centre of the 46B8-binding site, is highly conserved currently, a mutation to a phenylalanine could allow viruses to evade neutralisation without compromising their fitness [66]. Park and colleagues were also able to determine that the presence of stem-targeting antibodies may result in mutant virus selection to escape the immune response. Within human challenge models, a mutant virus of A338V H1N1 2009 was preferentially selected for in participants with higher levels of pre-challenge anti-HA stem antibodies [34].

### 6.2. Immunogenicity of bnAbs

BnMAbs and universal influenza vaccines in current clinical trials predominantly focus on antibodies targeting the HA stem. However, stem-targeting antibodies have demonstrated reduced immunogenicity compared to head-targeting antibodies [15]. The exact mechanisms behind this difference are not yet fully understood, although several explanations have been proposed.

Andrews and colleagues demonstrated that neutralising stem antibodies exhibited weaker binding to whole influenza virions compared to head-targeting antibodies, despite showing equivalent binding to recombinant HA [123]. One proposed explanation for this is that stem-targeting antibodies face steric hindrance, as their binding site on the HA stem may be located near the viral membrane, potentially limiting accessibility (Figure 2). Another proposed explanation is steric shielding of the stem epitopes by the head domain of the virion. The bulky head domain could prevent antibodies from binding to their epitopes present on the stem region of the trimer in vivo, reducing the immunogenicity of the stem-targeting antibodies. However, Harris and colleagues found that despite the close packing of HA trimers on the surface of a virion, 75% of HA trimers of the surface of an H1 virus were able to bind to the stem-targeting antibody C179 [124], suggesting that steric shielding may not play a role in the reduced immunogenicity of stem-targeting antibodies.

Additionally, the negative selection pressures exerted on B-cell clones producing stem-targeting antibodies may also play a role in their reduced immunogenicity. Stem-targeting antibodies are more prone to autoreactivity and polyreactivity, which could lead to immune tolerance or elimination of these B-cell clones. Andrews and colleagues identified that neutralising stem-targeting antibodies exhibited higher levels of polyreactivity compared to non-neutralising stem-targeting antibodies or head-targeting antibodies [123]. This observation was further supported by Bajic and colleagues, who tested 12 bnAbs—6 targeting the HA head and 6 targeting the stem. Of the stem-targeting antibodies, five displayed some degree of autoreactivity, whereas three head-targeting antibodies exhibited polyreactivity. Notably, CR6261, a stem-targeting bnMAb that had entered clinical trials for use as a therapeutic bnMAb (Table 3), showed both autoreactivity and binding to phospholipids, suggesting potential off-target interactions [125]. The presence of polyreactivity and autoreactivity in stem-targeting bnAbs raises concerns about their safety and efficacy as therapeutic agents. Polyreactivity may induce immune clearance mechanisms, such as B-cell anergy, and increase the risk of adverse effects due to off-target binding. These factors underscore the need for careful consideration of autoreactivity and polyreactivity in the development of bnMAbs, whether for therapeutic applications or as a foundation for universal influenza vaccines.

### 6.3. Antibody-Dependent Enhancement

The potential for antibody-dependent enhancement (ADE) of infection represents a critical concern in the development of bnAbs as therapeutic agents, or as a foundation for universal influenza vaccines. ADE occurs when virus-specific antibodies facilitate, rather than inhibit, viral infection. This phenomenon arises when antibodies are present at subtherapeutic concentrations or function in a non-neutralising manner, leading to increased disease severity. In such cases, antibodies may promote viral entry into host cells, thereby exacerbating infection, or contribute to the formation of excessive immune complexes, intensifying inflammatory responses [126,127,128,129].

Non-neutralising antibodies have been implicated in ADE during various viral infections, including the early stages of HIV-1 infection. Willey and colleagues demonstrated that, during the acute phase of HIV-1 infection, non-neutralising antibodies targeting the viral envelope can initiate complement-mediated ADE (C’-ADE), resulting in heightened levels of infection [126]. Similarly, studies on influenza have shown that certain non-neutralising antibodies can enhance disease outcomes. Winarski and colleagues reported that two non-neutralising mAbs that bind to the head domain of HA caused enhanced disease in mice after viral challenge with H3N2. Interactions of these mAbs with HA led to the destabilisation of the HA stem domain, resulting in increased viral load and pathogenicity in mice [130]. Historical data further support the role of non-neutralising antibodies in severe influenza outcomes. During the 2009 H1N1 pandemic, middle-aged individuals with pre-existing immunity to seasonal influenza exhibited higher rates of severe illness than expected. Monsalvo and colleagues identified the presence of cross-reactive, non-neutralising antibodies in this population, which were considered to be linked to an increase in immune-complex-mediated disease, with this population also having markers for complement activation via immune complexes. A similar mechanism was identified in fatal cases of the 1957 influenza pandemic [127]. These observations highlight the potential risks associated with non-neutralising antibodies during pandemic influenza cases.

In light of these concerns, the potential for bnAbs to trigger ADE is being carefully studied due to the ongoing clinical trials that are investigating the use of bnAbs as a therapeutic intervention following influenza infection. Rao and colleagues examined the human monoclonal IgG1 bnMAb MHAB5553A, which targets a conserved epitope in the vestigial esterase domain of HA and exhibits strong antiviral activity against multiple influenza B strains in murine models [66]. Notably, MHAB5553A did not enhance pathogenesis in female DBA/2J mice within this study [129].

Future research should explore strategies to mitigate ADE. One mitigation strategy involves blocking or preventing the binding of the antigen–antibody complex to Fc receptors. Modifications to the Fc region may reduce the risk of ADE. Wang and colleagues demonstrated this by introducing an LALA mutation into the Fc region of a SARS-CoV-2 mAB, eliminating ADE activity [131]. Another mitigation strategy is to use anti-complement-receptor antibodies. Complement receptor 3 (CR3) has been implicated in C’-ADE during HIV-1 and Flavivirus infection. Studies demonstrate that anti-CR3 antibodies can prevent C’-ADE-mediated infection for both of these viruses [132,133].

In conclusion, while bnAbs hold promise for influenza therapy and vaccine development, the risk of ADE must be thoroughly evaluated to ensure the safety and efficacy of these approaches, due to the risk of increased disease severity and possible link with fatal pandemic cases of influenza. Ongoing research into the mechanisms of ADE in the context of bnAbs will be crucial for the advancement of universal influenza vaccines and antibody-based therapies.

## 7. BnAbs in Current and Future Directions

Many bnAbs, and their respective epitopes, have been identified for the hemagglutinin protein of influenza, with many showing the ability to neutralise multiple influenza strains. These bnAbs provide an explanation for the ability for humans to tolerate mildly mutated strains, along with providing a possible route forward for the development of new therapeutics and influenza vaccines. However, limitations of these bnAbs are still present and must not be forgotten when moving forward with potential therapeutics. Further research into the limitations of bnAbs is required, to account for greater strain coverage, improve antibody potency, reduce the risk of viral escape, and address the potential for ADE. Exploring additional epitopes could enhance efficacy and lead to the discovery of new bnAbs with broader strain coverage. This approach may also enable combination therapies using multiple antibodies, providing better protection and reducing the risk of viral escape by requiring multiple simultaneous mutations.

Continued research and innovation will prove essential to realising the full potential of bnAbs and their associated epitopes, charting the way for effective and durable solutions to combat influenza on a global scale.

## Figures and Tables

**Figure 1 antibodies-14-00004-f001:**
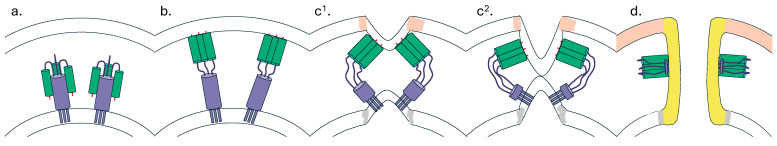
Steps in hemagglutinin endosomal collapse: The process begins when hemagglutinin (HA) binds to sialic acid on the host cell surface, facilitating viral endocytosis. In the endosome, HA initially exists in its (**a**) prefusion state and is bound to sialic acid. Upon endosomal acidification, HA releases the bound sialic acid and undergoes a conformational change to its (**b**) extended intermediate state, allowing the fusion peptide (shown in red) to insert into the endosomal membrane (shown in orange). The (**c**) extended intermediate state collapses, and HA undergoes a “jackknife” motion (**c^1^**) → (**c^2^**), drawing the viral membrane (shown in grey) and endosomal membrane together. As membrane fusion occurs, HA adopts its (**d**) post-fusion state, forming a fusion pore (shown in yellow) that facilitates the release of viral genetic material into the cytoplasm.

**Figure 2 antibodies-14-00004-f002:**
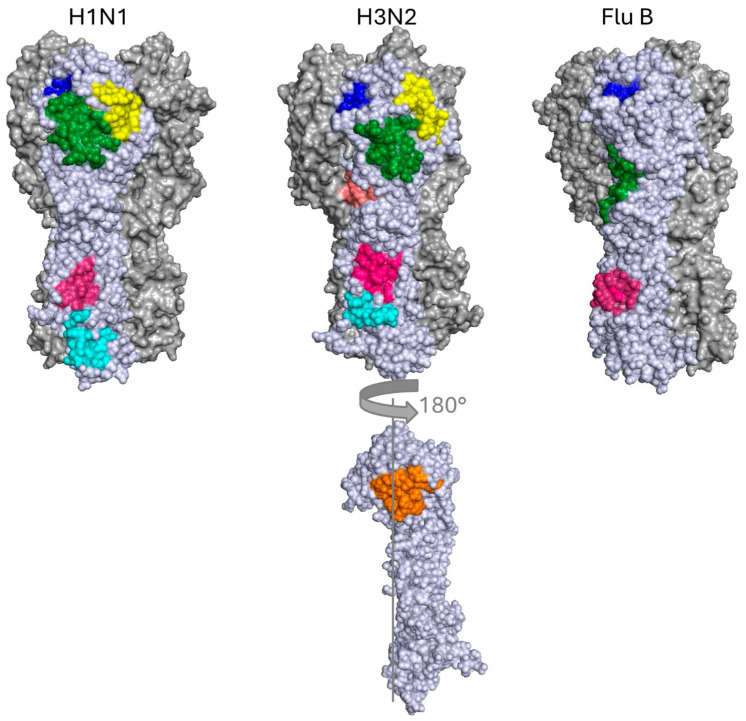
Approximate locations of stem epitopes in representative influenza A group 1 (A/South Carolina/1/1918(H1N1), PDB: 1RUZ), influenza A group 2 (A/Hong Kong/1/1968(H3N2), PDB: 4WE4), and influenza B (B/Hong Kong/8/73, PDB: 3BT6). The central stem (CS) epitope (pink) and fusion peptide (cyan H3N2) or fusion peptide and anchor epitope (cyan H1N1) are in the stem. Conversely, the RBS (blue), VE (green), and lateral patch (yellow) are situated in the head domain. The occluded epitope and the interface epitope (orange) are marked in orange on a single rotated representative H3N2 monomer.

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
