# Peer review of "How Broadly Neutralising Antibodies Are Redefining Immunity to Influenza"

_2073-4468, 2025, doi:10.3390/antib14010004_

Round 1

Reviewer 1 Report

Comments and Suggestions for Authors

It is well known that influenza virus infections cause significant morbidity and mortality worldwide. Rebecca et al discuss very well in this review the role of broadly neutralizing antibodies (bnAbs) in influenza infection. The review presents and highlights in detail the targeting of bnAbs against conserved regions of the viral hemagglutinin (HA) protein, which provides an important insight into new vaccine designs and platforms. Another interesting perspective is to show the potential of monoclonal antibodies (mabs) as a therapy against influenza, but also their limitations and the need for further research. I have some minor comments, with point 3 in need of proper revision:

1. In Figure 1, the words: Endosomal membrane, fused membrane, and viral membrane are impossible to read, perhaps the words should be replaced by symbols or colors and referenced in the legend.

2. In Table 2, in vitro protection should follow the same pattern as used for  neutralizing and binding words, that is, the word protection should be under in vivo.

3. In Table 3, check the spaces between numbers and letters

Author Response

We would like to thank Reviewer 1 for their very useful comments. In this regard we have made several changes outlined below.

Comment 1: In Figure 1, the words: Endosomal membrane, fused membrane, and viral membrane are impossible to read, perhaps the words should be replaced by symbols or colors and referenced in the legend.

Response 1: Thank you for the suggestion. Figure 1 has been changed to colour code the endosomal and viral membrane. These colours have been referenced in the figure legend. The words “endosomal membrane” and “viral membrane” have been removed.

Comment 2: In Table 2, in vitro protection should follow the same pattern as used for neutralizing and binding words, that is, the word protection should be under in vivo.

Response 2: Thank you for highlighting this omission. We corrected the title ‘in vivo protection’ within Table 2 to follow the same pattern as the other headers.

Comment 3: In Table 3, check the spaces between numbers and letters

Response 3: We have now corrected the spacing between the numbers and letters within Table 3.

We hope that this addresses their comments regarding the manuscript.

Reviewer 2 Report

Comments and Suggestions for Authors

The manuscript “How broadly neutralizing antibodies are redefining Immunity to influenza” by Steventon et al provides a comprehensive overview of bNAbs in the context of influenza virus immunity and vaccine development. While the review article focuses on the significant aspect of the bNAbs and present clinical trial data, but lack some novelty particularly considering recent review by Sun et al, 2024.  To ensure its impact, the authors should consider integrating new research findings, providing in-depth analyses of clinical trial results, and discussing innovative approaches that have emerged in recent years.

There are few points, which could be introduce to make this article more interesting to the broader audience in the field of immunology and virology.

1.       The brief mention of the challenges faced in the current vaccine strategies, particularly regarding antigenic drift and shift. More detailed discussion of these challenges could help emphasize the importance of bNAbs in overcoming these issues.

2.       To include more detailed comparative data on the efficacy of bnAbs targeting the HA stem versus those targeting the HA head. This could clarify why certain bnAbs are prioritized in research.

3.       In section 4 about clinical trials, provide valuable insights into the ongoing research surrounding bnAbs. However, the authors could enhance its evaluation by including more detailed results from these studies, such as patient demographics, dosage variations, efficacy etc.

4.       The paper briefly touches on antibody-dependent enhancement (ADE) as a potential concern with bNAbs. Authors could expand on the implications of ADE and how this risk may shape future clinical applications of bnAbs. Including a few potential strategies for overcoming the limitations discussed could provide a more comprehensive view.

5.       The paper could conclude with a more robust discussion on future research directions, including potential combination therapies (e.g., bnAbs with adjuvanted vaccines), novel delivery methods for vaccines, and the exploration of additional epitopes beyond those currently targeted.

6.       Authors can include a section describing the methods use to map new epitopes by neutralizing antibodies structurally. In HCV e.g. Legobodies or fiducial markers used in cryoEM described in recent studies. Shahid et al, 2021, JMB; Wu et al, 2021, PNAS; Mishra et al, 2023, Structure.

7.       There are a few older references that may benefit from being replaced with more recent studies to reflect the latest advancements in the field.

8.       It might be helpful to define acronyms (e.g., bnAbs) the first time they are used in the text, even though they are defined in the abstract.

Author Response

We would also like to thank Reviewer 2 for their comprehensive comments and critique. To address their comments we have made the following changes.

Comment 1: The brief mention of the challenges faced in the current vaccine strategies, particularly regarding antigenic drift and shift. More detailed discussion of these challenges could help emphasize the importance of bNAbs in overcoming these issues.

Response 1: Thank you for highlighting this omission. We have now added additional information on the challenges of predicting and the dangers of mismatching the correct antigen composition.

Comment 2: To include more detailed comparative data on the efficacy of bnAbs targeting the HA stem versus those targeting the HA head. This could clarify why certain bnAbs are prioritized in research.

Response 2: Information on mechanistic insights on the discrepancy between head and stem targeting antibodies has now been added. Thank you for highlighting the omission.

However, whilst an intriguing and potentially impactful suggestion, we felt that including direct “comparative data” such as IC50 or Kd was not objectively feasible. This is because the studies referenced in the review have tested their reported bnAbs against different HAs, often using different methods. In addition to this, some antibodies have only been tested against one viral strain (these are more likely to report very potent neutralization), while others have compared within groups or even between strains (which are more likely to have lower reported IC50 or Kd).

Furthermore, head antibodies have been well reported in the literature to be more potent than stem antibodies. However, to address the reviewer’s useful comments, we added additional mechanistic insights to the review text to highlight why this may be.

Comment 3: In section 4 about clinical trials, provide valuable insights into the ongoing research surrounding bnAbs. However, the authors could enhance its evaluation by including more detailed results from these studies, such as patient demographics, dosage variations, efficacy etc.

Response 3: Thank you for highlighting this. More details of the patient demographics and results of the clinical trials have now been included in section 4.

Comment 4: The paper briefly touches on antibody-dependent enhancement (ADE) as a potential concern with bNAbs. Authors could expand on the implications of ADE and how this risk may shape future clinical applications of bnAbs. Including a few potential strategies for overcoming the limitations discussed could provide a more comprehensive view.

Response 4: Thank you. A discussion on the potential strategies to overcome ADE has now been included.

Comment 5: The paper could conclude with a more robust discussion on future research directions, including potential combination therapies (e.g., bnAbs with adjuvanted vaccines), novel delivery methods for vaccines, and the exploration of additional epitopes beyond those currently targeted.

Response 5: A final discussion section has now been included, focusing on the future of this area of research. Thank you for highlighting this key omission in the original manuscript.

Comment 6: Authors can include a section describing the methods use to map new epitopes by neutralizing antibodies structurally. In HCV e.g. Legobodies or fiducial markers used in cryoEM described in recent studies. Shahid et al, 2021, JMB; Wu et al, 2021, PNAS; Mishra et al, 2023, Structure.

Response 6: In this instance, we felt that including a section describing methods for the mapping of new epitopes was beyond the scope of this review.

Comment 7: There are a few older references that may benefit from being replaced with more recent studies to reflect the latest advancements in the field.

Response 7: Thank you for highlighting this omission. We have review and updated some of the references. However, some older references were included in this review to reference the initial discoveries/original papers associated with bnAbs discovery.

Comment 8: It might be helpful to define acronyms (e.g., bnAbs) the first time they are used in the text, even though they are defined in the abstract.

Response 8: We have now done this. Thank you.

Reviewer 3 Report

Comments and Suggestions for Authors

How Broadly Neutralizing Antibodies are Redefining Immun-2 ity to Influenza

 Rebecca Steventon1,†, Lucas Stolle1,† and Craig Peter Thompson1,*

Steventon et al. made a detailed description regarding the neutralizing antibodies and their role in immunity to influenza.

·         Authors can explain on “All the antibodies of the central stem epitope are working on same principle?” if no, please mention how they are different ?

·         Please mention the antigenic site in this sentence otherwise it’s little confusing to the readers. “A second antigenic site has been identified below the CS epitope and closer to the 124 viral membrane (Fig 2, Table 1).”

·         Section 2.2 , line 127 : What is the relevant site in this “However, recently Group 1 influenza has also been 127 found to contain a relevant site near the viral membrane “

·         Line 143:  bnAbs against the HA Head Domain What are "sterically occluded antigenic sites" please explain?

·         Regarding Grouping of Epitopes: Authors discussing the stem and head domains to the specific epitopes (RBS, Lateral Patch, VE, and Occluded Site) is not in the flow with manuscript. Authors can add a sentence that introduces this section more smoothly, such as: Many broadly neutralizing antibodies (bnAbs) targeting the head domain have been identified, and their epitopes can be categorized into four distinct regions: the Receptor Binding Site (RBS), the Lateral Patch, the Vestigial Esterase (VE), and the Occluded Site (Fig. 2). So it will be clear to the readers about the transition of the domain specific antibodies.

·         Regarding Vestigial Esterase (VE), Authors mention that ADCC and crosslinking, but it would be helpful to explain why these mechanisms are important for in vivo protection. Authors can explain why VE-specific antibodies typically lack HAI activity, their ability to induce ADCC and crosslink HA trimers plays a key role in neutralizing the virus, potentially by promoting immune cell-mediated killing of infected cells or disrupting viral particle assembly

·         bnAbs in Clinical Trials: From Line 10-37 : The discussion of individual clinical trials for bnMAbs is valuable but could be structured more clearly. Breaking it up into distinct sub-sections (e.g., CT-P27, VIS410, MHAA4549A, etc.) would improve readability and help the reader track the different trials and their results. Additionally, consider summarizing each trial’s key findings in one or two sentences before delving into more detailed data.

·         Further Discussion on bnMAb Limitations: In this section

·         While the section briefly touches on limitations of bnMAbs (e.g., lack of significant efficacy in some trials, inactivation of CT-P27), a more thorough discussion of these challenges will be helpful for the readers to understand.

·         IgG Subclasses and Fc Receptor Interactions: Authors should clearly explain how IgG3 might overcome some of the limitations observed with stem-targeting bnMAbs, such as CR8020, and why this is important for improving therapeutic efficacy.

·         Immunogenecity of the neutralizing antibodies is crucial, especially in the context of long-term therapy or vaccination. You mention that stem-targeting antibodies demonstrate reduced immunogenicity compared to head-targeting antibodies, but the reasons for this difference could be explored in more detail. For example, you could mention how the structural differences between the stem and head domains of HA might influence antibody recognition or immune response.

Author Response

We would like to thank Reviewer 3 for their insights and comments. In response to Reviewer’s 3 comments, the following revisions have been made.

Comment 1: Authors can explain on “All the antibodies of the central stem epitope are working on same principle?” if no, please mention how they are different?

Response 1: Thank you for highlighting this. We have now rephrased section 2.1 to more clearly explain the difference between different central stalk antibodies.

Comment 2: Please mention the antigenic site in this sentence otherwise it’s little confusing to the readers. “A second antigenic site has been identified below the CS epitope and closer to the 124 viral membrane (Fig 2, Table 1).”

Response 2: We have now referenced the antigenic site in this sentence. Thank you.

Comment 3: Section 2.2 , line 127 : What is the relevant site in this “However, recently Group 1 influenza has also been 127 found to contain a relevant site near the viral membrane “

Response 3: Thanks for highlighting this. The relevant antigenic site for Group 1 has now been mentioned.

Comment 4: Line 143:  bnAbs against the HA Head Domain What are "sterically occluded antigenic sites" please explain?

Response 4: A clearer explanation for “sterically occluded” has now been provided.

Comment 5: Regarding Grouping of Epitopes: Authors discussing the stem and head domains to the specific epitopes (RBS, Lateral Patch, VE, and Occluded Site) is not in the flow with manuscript. Authors can add a sentence that introduces this section more smoothly, such as: Many broadly neutralizing antibodies (bnAbs) targeting the head domain have been identified, and their epitopes can be categorized into four distinct regions: the Receptor Binding Site (RBS), the Lateral Patch, the Vestigial Esterase (VE), and the Occluded Site (Fig. 2). So it will be clear to the readers about the transition of the domain specific antibodies.

Response 5: The recommended sentence structure for introducing the head epitopes has now been adopted. Thank you for the suggestion.

Comment 6: Regarding Vestigial Esterase (VE), Authors mention that ADCC and crosslinking, but it would be helpful to explain why these mechanisms are important for in vivo protection. Authors can explain why VE-specific antibodies typically lack HAI activity, their ability to induce ADCC and crosslink HA trimers plays a key role in neutralizing the virus, potentially by promoting immune cell-mediated killing of infected cells or disrupting viral particle assembly

Response 6: More information on why the VE lacks HI activity, as well as why and how these antibodies display ADCC has now been provided. Thank you.

Comment 7: bnAbs in Clinical Trials: From Line 10-37 : The discussion of individual clinical trials for bnMAbs is valuable but could be structured more clearly. Breaking it up into distinct sub-sections (e.g., CT-P27, VIS410, MHAA4549A, etc.) would improve readability and help the reader track the different trials and their results. Additionally, consider summarizing each trial’s key findings in one or two sentences before delving into more detailed data.

Response 7: We have now restructured section 4 to break up the one big section into multiple distinct sections and including a summarisation of each bnMAb.

Comment 8: Further Discussion on bnMAb Limitations: In this section:

Comment 8.1: While the section briefly touches on limitations of bnMAbs (e.g., lack of significant efficacy in some trials, inactivation of CT-P27), a more thorough discussion of these challenges will be helpful for the readers to understand.

Response 8.1: Additional information has been given on clinical trials and their limitations. Thanks for highlighting this omission.

Comment 8.2: IgG Subclasses and Fc Receptor Interactions: Authors should clearly explain how IgG3 might overcome some of the limitations observed with stem-targeting bnMAbs, such as CR8020, and why this is important for improving therapeutic efficacy.

Response 8.2: Further discussion has been included on the limitations of bnMAbs and the potential to overcome these with IgG subclass switched and Fc receptor interactions.

Comment 8.3: Immunogenicity of the neutralizing antibodies is crucial, especially in the context of long-term therapy or vaccination. You mention that stem-targeting antibodies demonstrate reduced immunogenicity compared to head-targeting antibodies, but the reasons for this difference could be explored in more detail. For example, you could mention how the structural differences between the stem and head domains of HA might influence antibody recognition or immune response.

Response 8.3: Further discussion on the immunogenicity difference between stem and head targeting antibodies has been included.

Thank you for the very useful suggestions that have improved the manuscript.

Round 2

Reviewer 2 Report

Comments and Suggestions for Authors

Dear Authors,

Thank you for your thorough revisions and detailed responses to the comments.  I have no further comments or suggestions, and I am confident that this version is suitable for publication.

Congratulations on your excellent work, and I wish you success with your research.

Warm regards